# Novel CuMgAlTi-LDH Photocatalyst for Efficient Degradation of Microplastics under Visible Light Irradiation

**DOI:** 10.3390/polym15102347

**Published:** 2023-05-17

**Authors:** Shengyun Jiang, Mingshan Yin, Huixue Ren, Yaping Qin, Weiliang Wang, Quanyong Wang, Xuemei Li

**Affiliations:** 1School of Municipal and Environmental Engineering, Shandong Jianzhu University, Jinan 250101, China; 2020040215@stu.sdjzu.edu.cn (S.J.);; 2Beicheng Environmental Engineering Co., Ltd., Jinan 250101, China; 3Shandong Huacheng Urban Construction Design Engineering Co., Ltd., Jinan 250101, China

**Keywords:** microplastics, photocatalysis, polystyrene, polyethylene, layered double hydroxide

## Abstract

Microplastics (MPs) in the water system could easily enter the human body and pose a potential threat, so finding a green and effective solution remains a great challenge. At present, the advanced oxidation technology represented by photocatalysis has been proven to be effective in the removal of organic pollutants, making it a feasible method to solve the problem of MP pollution. In this study, the photocatalytic degradation of typical MP polystyrene (PS) and polyethylene (PE) by a new quaternary layered double hydroxide composite photomaterial CuMgAlTi-R400 was tested under visible light irradiation. After 300 h of visible light irradiation, the average particle size of PS decreased by 54.2% compared with the initial average particle size. The smaller the particle size, the higher the degradation efficiency. The degradation pathway and mechanism of MPs were also studied by GC–MS, which showed that PS and PE produced hydroxyl and carbonyl intermediates in the process of photodegradation. This study demonstrated a green, economical, and effective strategy for the control of MPs in water.

## 1. Introduction

As a kind of synthetic organic polymer, plastic is widely used in various industries because of its lightweight, high strength, durability, and low cost [1]. However, it also causes serious environmental pollution problems. When waste plastics enter the environmental system, they are decomposed into smaller plastic fragments, microplastics (MPs), through various actions, such as weathering and erosion [2,3]. The concept of MPs was first put forward by Thompson et al. [4] in 2004 and has attracted wide attention as a new pollutant. MPs are defined as plastic fragments with a particle size smaller than 5 mm, which could be divided into primary MPs and secondary MPs in accordance with their sources [5]. The widely used ones are mainly polyethylene (PE), polystyrene (PS), and polypropylene (PP).

As a highly mobile environmental medium, water is not only an important carrier of MPs migration but also one of the main gathering locations of MPs [6]. In addition, MPs existing in water bodies are ideal carriers for many pollutants because of their large specific surface area and they are easy to enrich heavy metals in water. After being eaten by aquatic organisms, MPs could enter the human body through the food chain and threaten human health [7]. Nowadays, MPs have been found in human feces [8]. Plastic particles less than 130 µm in diameter may trigger local immune responses through transfer into human tissues [9]. Urban waters are closely related to people’s daily life and have a far-reaching effect on human health [6]. Therefore, in this context, removing MPs from urban waters is urgent.

Nowadays, many methods, such as coagulation [10], filtration [11], biodegradation [12], and photocatalysis [13], have been used to remove MPs. Although the removal rates of traditional methods, such as coagulation and filtration, are very high at even up to 99%, MPs still exist in the sludge and are easy to return to the environment [14]. Biodegradation becomes complex because of its long degradation time, which requires biological separation and cloning of degradable enzymes. Advanced oxidation processes (AOPs) can generate strong oxidizing radicals such as hydroxyl radicals (^•^OH) and sulfate radicals (SO_4_^•−^) to completely mineralize organic pollutants into H_2_O and CO_2_, so as to achieve complete removal of pollutants. Depending on the mode of radical generation and reaction conditions, AOPs include persulfate oxidation, photocatalytic oxidation, ozone oxidation, and Fenton oxidation [15]. Photocatalytic oxidation, as one of the AOPs, has the advantages of using solar energy as a clean energy source, high degradation efficiency, and the generation of harmless byproducts. It has been applied to the removal of MPs in recent years. Nabi et al. [13] found that the degradation rate of 5 µm PS microspheres was 99.99% after UV irradiation with 254 nm for 24 h. Ariza et al. [16] found that the degradation effect of N-TiO_2_ was better than that of sol–gel N-TiO_2_ (6.40% vs. 2.86%). When TiO_2_ or a similar semiconductor is excited by a light source with energy greater than its intrinsic band gap, holes (h^+^) and electrons (e^−^) are generated in the valence band (VB) and conduction band (CB), respectively, and both h^+^ and e^−^ react with H_2_O, OH^−^ and O_2_ adsorbed on the surface of the semiconductor to generate reactive oxygen species (ROS), such as ^•^OH and superoxide radicals (^•^O_2_^−^). These species initiate the polymer degradation process leading to chain breaking and complete mineralization to H_2_O and CO_2_ [6]. The photocatalytic process is described by the following equation:(1)$$\mathrm{Photocatalyst}\overset{\mathrm{hv}}{\to }h^{+}{+e}^{-}$$
(2)$$h^{+}+H_{2}O\to {}^{•}OH$$
(3)$$e^{-}{+O}_{2}\to {}^{•}O_{2}^{-}$$
(4)$$\mathrm{Organic}\;\mathrm{Pollutant}+\mathrm{ROS}\to {\mathrm{CO}}_{2}+H_{2}O$$

However, due to the rapid recombination of hole electron pairs and high bandgap values of most current photocatalysts, the current photocatalytic degradation performance is not ideal (such as low degradation efficiency, response only under UV light, and long irradiation time), thus hindering further applications [6,17]. Therefore, exploring high-efficiency photocatalysts with enhanced performance and capable of degrading under visible light is the development direction of photocatalytic technology in the future.

Layered double hydroxides (LDHs), also known as hydrotalcite materials, are inorganic compounds mainly composed of divalent and trivalent metal cations. The general formula is as follows: ${\left[M_{1-x}^{2+}M_{x}^{3+}{\left(OH\right)}_{2}\right]}^{x+}{\left[A^{n-}\right]}_{x/n}\cdot mH_{2}O$, where M^2+^ is the divalent cation (such as Mg^2+^ or Zn^2+^), M^3+^ is the trivalent cation (such as Al^3+^ or Fe^3+^), A^n−^ is the exchangeable anion (usually CO_3_^2−^, Cl^−^, or NO_3_^−^), and x is usually 0.2 < x < 0.33 [18]. Due to their excellent adsorption performance and photocatalytic performance, LDHs have received great attention in recent years. Moreover, because the motherboard cations have ion adjustability, the material structure could be functionally adjusted in accordance with the type of pollutants to achieve maximum degradation performance [19]. The current literature that reports on the use of LDH to remove MPs mainly focuses on its adsorption performance on MPs [20,21], but no in-depth study on the photocatalytic degradation efficiency and performance of MPs is available.

In this study, a novel and efficient quaternary MgAl-based quaternary LDH composite photocatalyst was prepared by regulating metal cations via the octahedral deformation principle [22]. By taking the bivalent and trivalent metals (Mg and Al) of Type I as the basic framework, the tetravalent metal (Ti) with photocatalytic activity was introduced, and coupled hybridizable Type II bivalent metal (Cu) was involved in the construction of motherboards to improve the photocatalytic activity. The CuMgAlTi-R400 photocatalyst was prepared for the degradation of MPs PS and PE. The degradation pathway and mechanism of PS and PE removal by CuMgAlTi-R400 were revealed in-depth, and good results were obtained. This work not only opens up a new avenue for the use of photocatalytic technology to remove MPs but also provides a new idea for the complete removal of MPs in water in a low-cost and efficient manner.

## 2. Materials and Methods

### 2.1. Materials

The PE and PS used in the experiment were purchased from Huachuang Plastic Raw Material Company (Guangzhou, China). CuCl_2_·2H_2_O, MgCl_2_·6H_2_O, AlCl_3_·6H_2_O, TiCl_4_, NaOH, DMPO, CH_3_OH, and Na_2_CO_3_ were purchased from Aladdin Chemical Reagent Company (Shanghai, China). All chemicals used were analytically pure and could be used without further purification. Ultrapure (UP) water was always used in this study.

### 2.2. Synthesis of CuMgAlTi-R400

In this experiment, a carbonate intercalation quaternary Mg-Al-based LDH photocatalyst was prepared by the coprecipitation method. The four metal salts CuCl_2_·2H_2_O (0.001 mol), MgCl_2_·6H_2_O (0.03 mol), AlCl_3_·6H_2_O (0.01 mol) and TiCl_4_ (0.009 mol) were dissolved in 50 mL of ultrapure water to form a salt solution. In addition, NaOH (0.08 mol) and Na_2_CO_3_ (0.008 mol) were dissolved in 50 mL of ultrapure water to form an alkaline solution. The alkali solution was slowly dropped into the salt solution and stirred continuously until the pH of the mixed solution reached 9. The mixture was transferred to an oil bath and crystallized at 60 °C for 12 h. After the crystallization was completed, the mixture was left to cool and then filtered, and the filter cake was dried and then ground to obtain a quaternary LDH photocatalyst precursor. The precursor was calcined and activated by a muffle furnace (SX-4-10p, Tianjin Tester Instrument Co., Ltd. (Tianjin, China)). The calcination temperature was 400 °C, and the calcination time was 4 h. After the powder was cooled, hydration reconstruction was carried out on the basis of the “memory effect” of the LDH material, and the powder was placed in UP water and stirred for 12 h to rebuild its layered structure. Finally, the powder was dried to obtain the final quaternary LDH photocatalyst, denoted as CuMgAlTi-R400.

### 2.3. Photodegradation Experiments

All photocatalysis tests were carried out in a closed reaction chamber. A 2000 mesh stainless steel screen (Hebei Screen Products Factory, Shijiazhuang, China) was folded into a funnel and placed in the middle and upper part of the beaker to reduce the effect of powder photocatalyst-coated MPs on the degradation and subsequent separation of MPs. The process is then to weigh 200 mg of MP into a 250 mL beaker, weigh an equal amount of photocatalyst into a sieve, add 200 mL of ultrapure water, and put it into the photocatalytic reaction box. Using a Xe lamp (50 W, λ = 320–780 nm, PLS-SXE300+, Beijing Perfectlight Technology Co., Ltd., Beijing, China) as a light source to provide a 30 mW/cm^2^ irradiance visible light. The visible light catalytic reaction was carried out under magnetic stirring at 298 k and 600 r/min, and the samples were sampled and tested after a period of time.

The reacted solution was passed through the 0.45 μm organic filter membrane (Tianjin Branch Lung Experimental Equipment Co., Ltd., Tianjin, China), the MP particles were extracted and retained after the filter membrane was dried, and the filtrate was stored in the refrigerator. In addition, the particle size of the MPs was measured using an optical microscope (Olympus, BX53, Tokyo, Japan) connected to a digital camera that captured the image. The image analysis software ImageJ (version 2.9.0) was used to determine the size distribution. On the basis of the obtained particle size, the percentage of particle size reduction was calculated as follows:(5)$$\mathrm{Particle}\;\mathrm{size}\;\mathrm{reduction}\;\mathrm{percentage}\left(\%\right)\;=\;(\frac{\mathrm{Initial}\;\mathrm{size}\;-\;\mathrm{Final}\;\mathrm{size}}{\mathrm{Initial}\;\mathrm{size}})\;\times \;100$$

### 2.4. Characterization

The Bruker D8 Advance X-ray diffractometer (XRD) was used to characterize the phase of the sample by scanning CuMgAlTi-R400 at a scanning rate of 5°/min in the range of 5°–70°. The UV–vis absorption spectrum was recorded by the Shimadzu UV2700 spectrophotometer.

The infrared spectra of the samples in the range of 4000–450 cm^−1^ were recorded by the Bruker TENSORII infrared spectrometer, which was used to qualitatively observe the molecular changes in functional groups. CuMgAlTi-R400 used Fourier transform infrared (FTIR)-IR, and MPs used FTIR-ATR. The effect of photocatalytic oxidation of MPs was monitored by measuring the carbonyl index (CI), that is, the ratio of the peak area of the carbonyl part of the infrared spectrum to the reference peak area. For different kinds of MPs, the characteristic peak with slight change with an aging degree is usually selected as the reference peak. For oxidized MPs, the carbonyl is detected in the wide infrared region of 1550–1850 cm^−1^. In this study, the CI of PS was the ratio of the area under the absorption peak of 1710 and 1410 cm^−1^, and the CI of polyethylene was the ratio of the area under the absorption peak of 1730 and 2850 cm^−1^.

The morphology and structure of the samples were analyzed by a scanning electron microscope (Zeiss Gemini SEM 300) and a microscope (Olympus, BX53).

Thermo Scientific K-Alpha X-ray photoelectron spectroscopy (XPS,) was used to measure the VB values of the semiconductor photocatalysts. The active oxidants produced in the photocatalytic system were measured by Bruker EMXnano Electron Paramagnetic Resonance (EPR), using ultrapure H_2_O or CH_3_OH as the reaction solution, and DMPO as the trapping agent to capture ^•^OH and ^•^O_2_^−^.

The Agilent GC8890-5977B (America) gas chromatography–mass spectrometry (GC–MS) was used to analyze and identify the organic intermediates formed during the degradation of MPs.

## 3. Results

### 3.1. MP Characterization

Figure 1a shows the FTIR spectrum of PS, which showed the infrared spectrum of PS before photodegradation. The bands at 2972 and 2882 cm^−1^ are related to the stretching vibration of methylene C-H_2_ [23]. The characteristic band of the benzene ring appears around 1600 cm^−1^ (1539, 1650, and 1695 cm^−1^), which is related to the stretching mode of the C=C bond. The band between 450 and 800 cm^−1^ represents the out-of-plane bending mode of the C-H bond in the aromatic ring [24]. The stretching of C≡C is shown at 2360 cm^−1^. Meanwhile, the peaks at 3670 and 3622 cm^−1^ are related to hydroxyl or hydroperoxyl [25], which may be produced during preparation or storage.

Figure 1b illustrates the FTIR spectrum of PE. The bands at 2925, 2846, 1472, and 718 cm^−1^ are attributed to the alkyl chain of PE [26]. The bands at 2925 and 2846 cm^−1^ belong to C-H_2_ asymmetrical and symmetrical stretching vibrations, and the 1472 and 718 cm^−1^ bands are related to C=C stretching and (C-H_2_)_n_ rocking vibrations [27].

Figure 1c,d show the undegraded scanning electron microscopic (SEM) image of PS and PE, respectively. The average diameter of PS was 39.0 ± 5.6 μm, and that of PE was 25.9 ± 3.6 μm. The sphere was intact, and the surface was dense.

### 3.2. Structure and Characterization of CuMgAlTi-R400 Photocatalysts

Figure 2a illustrates the X-ray diffraction spectrum of CuMgAlTi-R400. The results showed that characteristic crystal planes (003), (006), (012), (110), and (113) were observed at 2θ = 11.39°, 23.1°, 25.31°, 34.54°, 60.85°, and 62.12°, respectively. These are the characteristic diffraction peaks of LDHs, indicating that the synthesized materials formed a good layered structure [28]. Figure 2a also shows two small peaks at 25.31° and 36.41°, which are the characteristic diffraction peaks of the typical crystal plane (101) of anatase and rutile TiO_2_, respectively. Thus, the Ti element could be assumed to partly participate in the construction of the main laminate as Ti^4+^ metal cations and a small part could form TiO_2_ on the surface of the material, as previously reported [29]. Moreover, in the mixed-phase TiO_2_ nanocomposites, the two crystal forms have a synergistic effect [30], which could effectively improve the photocatalytic activity of the materials.

Figure 2b shows the FTIR spectrum of CuMgAlTi-R400. The broad absorption peak at 3437 cm^−1^ is caused by the stretching vibration of the O-H group in LDHs [31]. The weak absorption peak at 2976 cm^−1^ is attributed to the OH stretching mode of interlayer water molecules [32]. Two peaks could be observed near 1500 cm^−1^ (1638 and 1430 cm^−1^): the former is caused by the deformation vibration of H-O-H in the interlayer water molecules, and the latter belongs to the v_3_ mode of interlayer anion carbonate (CO_3_^2−^) [33]. The absorption peaks at 1160 and 1051 cm^−1^ could be attributed to the v_1_ mode of carbonate anions. The peak observed at 877 cm^−1^ could be attributed to the v_2_ mode of carbonate anions. Although this band is infrared inactive in free carbonates, it is activated due to the decrease in the symmetry of carbonate anions in the interlayer [32]. The two peaks in the range of 450–700 cm^−1^ are caused by the vibration of the metal–oxygen bond (M-O-M/O-M-O) on the main laminate of the layered material [34].

Figure 2c shows the absorption spectrum of CuMgAlTi-R400, which showed strong absorption in the UV region (200–400 nm). This finding could be attributed to the doping of Ti^4+^ ions. The CuMgAlTi-R400 also absorbs in the visible region at 400–500 nm and 600–800 nm, which is attributed to the doping of Cu^2+^. The classical Tauc method was used to estimate the so-called optical band gap as follows [35]:(6)$$\alpha h\nu =A{\left(h\nu -E_{g}\right)}^{n/2}$$
where *α*, *v*, *E_g_*, and *A* are absorption coefficient, optical frequency, bandgap energy, and constant, respectively. The parameter *n* depends on the type of optical transition in the semiconductor (*n* = 1 for direct transition and *n* = 4 for indirect transition).

As shown in Figure 2c, the bandgap of CuMgAlTi-R400 was 2.82 eV, which was significantly smaller than that of TiO_2_ (3.2 eV). CuMgAlTi-R400 showed excellent photocatalytic performance in the visible light region.

Figure 2d shows the SEM diagram of CuMgAlTi-R400. The materials reconstructed by calcined hydration showed an ordered sheet structure resembling petals, with a particle size of 200–300 nm, uniform size, and several pores between sheets, thus showing a large specific surface area, which could provide more active sites for photocatalytic degradation experiments and improve photocatalytic activity.

Considering the excellent properties of CuMgAlTi-R400 mentioned above, it was used in the photocatalysis experiment.

### 3.3. Photocatalytic Degradation Tests

#### 3.3.1. MP Morphology Analysis

The surface morphology of MPs before and after photodegradation was detected by SEM to explore the physical properties of MPs under visible light degradation. Figure 3 shows the SEM microphotos of PS and PE MPs after photodegradation in different time periods. The surface of the original MP particles was relatively smooth (Figure 3a,f,k,o). With the extension of degradation time, cracks and voids on the surface were gradually observed. As shown in Figure 3g,p, after 50 h of degradation, a large number of cracks began to appear on the surface of PS and PE, and then with the extension of time, the cracks gradually deepened and showed a cavity (Figure 3g–j,p–r), resulting in volume expansion and cracking into small particles (Figure 3a–e,k–n). The degradation effect of PS was obviously better than that of PE, which may be due to the large number of chromophoric groups in the preparation of PS [36]. The reason for this change in the surface of MPs may be due to the removal of volatile photodegradation byproducts and the increase in crystallization resulting in cracks and voids in the surface layer [13,37]. The formation of surface cracks and voids could increase the degradation effect because it provides a method for oxygen to penetrate deeper into the sample and enhance photo-oxidation. The reduction of volume is also more conducive to the degradation of MPs.

In addition, the particle size of MPs before and after degradation was measured. Table 1 and Table 2 summarize the particle size and average particle size distribution of PS and PE at different photodegradation times, respectively. With the increase in irradiation time, the particle size of MPs gradually decreased, indicating that the obvious fragmentation process of MPs occurred after photocatalytic degradation, consistent with the results shown in Figure 3, as caused by the degradation of the MP polymer chain.

The decrease in the percentage of MP particle size with photodegradation time is shown in Figure 4. In the process of photocatalytic degradation, the size of PS and PE decreased by 44.5% and 33.7%, respectively, after 200 h of visible light irradiation, and even the size of PS MPs decreased by more than 54% after 300 h of visible light irradiation. Among the many catalysts in Table 3 the performance is relatively excellent.

#### 3.3.2. FTIR Spectroscopic Analysis

FTIR-ATR was used to evaluate the relative oxidation of the MP surface after photodegradation (Figure 5 and Figure 6). The FTIR spectra of PS and PE MPs obtained by photodegradation under visible light at different time intervals were shown. By comparing the spectra obtained before and after photodegradation, the original bands of MPs were found to become more significant with the increase in degradation time after 50 h of photodegradation. The increase in the strength of the bands corresponding to C=C stretching and C≡C stretching indicated that chain or bond breakage (due to the breakage of the C-C or C-H bond) occurred [40]. The intensity of the PS band increased at first and then decreased with the increase in degradation time, which was mostly the characteristic peak of a benzene ring and may be related to the existence of a benzene ring. Figure 5b and Figure 6b show the bands corresponding to the stretching modes of hydroxyl and hydroperoxyl groups between 3300 and 3800 cm^−1^ increased significantly.

PS and PE formed new functional groups in 1600–1800 cm^−1^ (Figure 5d: 1710 cm^−1^; Figure 6d: 1730 cm^−1^) due to chemical changes, indicating the formation of carbonyl groups, consistent with previous studies [41]. With the increase in photodegradation time, the wide peak intensity at 1600–1800 cm^−1^ increased, which was formed by the overlapping combination of different tensile vibration bands of carboxylic acid, ketone, aldehyde, and lipid groups [42]. The increase in carbonyl groups in oxidized MPs was proportional to the number of chain breaks [8], consistent with the loss trend of MPs, as shown in Figure 4. The formation of hydroxyl and carbonyl groups occurred simultaneously in the process of photodegradation. The above observations showed that the increase in the intensity of functional groups, such as -C=O, -OH, and -OOH, indicated that the photo-oxidation mechanism of MPs occurred under visible light irradiation of CuMgAlTi-R400 [40].

Figure 5e and Figure 6e also show that the corresponding infrared bands of large-sized PS and PE were obviously weaker than those of small-sized MPs after photodegradation for 200 h. This finding showed that the smaller the particle size is, the higher the degradation efficiency under the same degradation conditions.

Given that the peak intensity in the infrared spectrum could not be directly used to quantify the number of functional groups, CI is most commonly used to roughly express the carbonyl content in MPs and evaluate the degradation of plastics. In the later stage of photodegradation, ketones, aldehydes, and carboxylic acids are formed. Although CI is an important tool for determining degradation, it does not take into account the stage of degradation or whether full mineralization occurs (CO_2_ and H_2_O) [27,43].

The changing trend of CI of MPs with degradation time is shown in Figure 5f and Figure 6f. With the increase in degradation time, the value of CI gradually increased, and after 200 h of visible light irradiation, the CI of PS and PE increased by 8.6 times and 2.7 times, respectively, compared with that of the original MPs, and PS increased by 14.3 times after 300 h of degradation. This finding indicated that ketones, aldehydes, or carboxylic acids may exist as intermediates formed before complete mineralization.

#### 3.3.3. GC–MS Analysis

The properties of the intermediates formed during the degradation of MPs were characterized by GC–MS. The chromatograms and mass spectra of PS and PE were obtained at 50 and 200 h, respectively, as shown in Figure 7. In Figure 7a,b, the fragment peak began to appear after 50 h of photodegradation and increased greatly at 200 h with the extension of photodegradation time. A large number of ion fragments were formed in the process of photodegradation of MPs. The *m*/*z* values of oxygen-containing intermediates were 43 (C_2_H_3_O^+^), 57 (C_3_H_5_O^+^), and 85 (C_5_H_9_O^+^), and the peak intensity was very good, indicating that the degradation intermediates were mainly oxidized compounds [13].

In various degradation intermediates of MPs, *m*/*z* = 71 belongs to butyraldehyde, and in various degradation intermediates of PS (Figure 7a), *m*/*z* values = 93, 119, 121, and 225 belong to phenol, acetophenone, benzoic acid, and benzoic anhydride, respectively [44]. This finding is consistent with the type of compounds produced by infrared spectrum analysis. As shown in Figure 7c,d, MPs produced more abundant organic compound intermediates with the extension of degradation time. Through the analysis of GC–MS, more intermediates, such as esters, aldehydes, ketones, and linear alkanes, were identified (Table 4 and Table 5). After further cracking, MPs finally transformed into water and CO_2_, which further confirmed the degradation mechanism of SEM analysis. These results suggested that photocatalysis using CuMgAlTi-R400 may be an effective alternative for the degradation of MPs because it could reduce polymers to shorter chains in a short time under visible light, ultimately fully mineralizing MPs into water and CO_2_.

#### 3.3.4. Reaction Mechanism

In order to investigate the degradation mechanism of photocatalytic degradation of MPs, the valence band position (E_VB_) of CuMgAlTi-R400 was measured by VB-XPS, and the result is shown in Figure 8a, and the E_VB-XPS_ was measured as 2.55 eV. The obtained potential was converted to the standard hydrogen potential E_VB-NHE_ of 2.31 eV by E_VB-NHE_ = φ (Instrumental function: 4.2 eV) + E_VB-XPS_ (2.55 eV) − 4.44, which is higher than E(OH^−^/^•^OH) = 1.99 eV, demonstrating that the semiconductor photocatalyst can oxidize OH^−^ in water to produce ^•^OH [45]. Furthermore, according to E_g_ = E_VB_ − E_CB_, E_CB_ = −0.51 eV, which is lower than E (O_2_/^•^O_2_^−^) = −0.33 eV and can produce ^•^O_2_^−^ [46]. Combined with the EPR patterns (Figure 8b,c) under dark conditions without the signals of ^•^OH and ^•^O_2_^−^, while a clear DMPO- ^•^OH and DMPO-^•^O_2_^−^ were observed after 120 s of visible light irradiation, indicating that ^•^OH and ^•^O_2_^−^ were the main active substances.

In general, the photodegradation process of MPs may take place in several steps, including chain initiation, adsorption of molecular oxygen, chain growth, secondary reaction, and termination [47]. The reaction mechanism diagram is shown in Figure 9. At the beginning of illumination, CuMgAlTi-R400 was excited by visible light, which caused electrons to flow from the valence band to the conduction band, producing h^+^ and e^−^ (Equation (7)), which could react with the OH^−^ and O_2_ adsorbed on the photocatalyst to form ^•^OH and ^•^O_2_^−^ (Equations (8) and (9)). The initiation step could be achieved by destroying the main chain of the polymer by ^•^OH, resulting in carbon-centered free radicals (Equation (10)) [25]. In the chain growth stage, these free radicals reacted with O_2_ to form peroxy radicals (Equation (11)), which produced hydroperoxides (ROOH) by seizing hydrogen atoms in the surrounding environment (Equation (12)) [5]. Free oxygen radicals (RO^•^) and hydroxyl radicals (^•^OH) were formed by the cleavage of weak O-O bonds (Equation (13)). RO^•^ continued to react through hydrogen capture, chain breaking, and rearrangement to produce inert intermediates, such as alcohols, aldehydes, and ketones [40]. Finally, these intermediates could continue the photocatalytic oxidation process until CO_2_ and H_2_O were formed (Equation (14)). In addition, the formation of free radicals in the photocatalytic process could promote the production of different intermediates, which further reacted with molecular oxygen and led to low molecular compounds, as confirmed by GC–MS analysis.
(7)$$\mathrm{CuMgAlTi}-R400+\mathrm{hv}⇌h_{\mathrm{VB}}^{+}+e_{\mathrm{CB}}^{-}$$
(8)$$h_{\mathrm{VB}}^{+}+OH^{-}\to {}^{•}\mathrm{OH}$$
(9)$$e_{\mathrm{CB}}^{-}+O_{2}\to {}^{•}O_{2}^{-}$$
(10)$$\mathrm{MPs}+{}^{•}OH\to R^{•}$$
(11)$$R^{•}+O_{2}\to ROO^{•}$$
(12)$${\mathrm{ROO}}^{•}+\mathrm{RH}\to \mathrm{ROOH}+R^{•}$$
(13)$$\mathrm{ROOH}\to RO^{•}+{}^{•}\mathrm{OH}$$
(14)$$RO^{•}\to \mathrm{Intermediates}\;\mathrm{products}\to CO_{2}+H_{2}O$$

## 4. Conclusions

In this work, visible light photocatalytic degradation of MPs was carried out for the first time by using LDH CuMgAlTi-R400. This study successfully proved the feasibility of using CuMgAlTi-R400 photocatalyst excited by visible light to degrade MPs in water. With the extension of visible light irradiation time, the photocatalytic oxidation of PS and PE led to the increase in CI of MPs, the chain breaking of polymer molecules; the production of ketones, aldehydes, esters, and other intermediates; and the loss of volatile intermediates resulted in wrinkles, cracks, and cavities on the plastic surface and then broken into smaller particles to achieve final degradation, which was confirmed by FTIR, SEM, GC–MS, and microscopy. In addition, VB-XPS and EPR tests showed that ^•^OH and ^•^O_2_^−^ are the main active species in the photocatalytic reaction system. The results showed that the average particle size of PS and PE decreased by 54.2% and 33.7%, respectively, after visible light irradiation for 300 and 200 h. Moreover, the smaller the particle size was, the higher the degradation efficiency. These results provide new insights into the use of clean technologies to solve global MP pollution and reduce byproducts.

The MPs in the actual water body are more complex and excellent carriers for pollutants, such as heavy metals. The excellent adsorption properties of LDHs could be used to realize the double removal of heavy metals and MPs. This content will be discussed in the next step of the authors’ group.

## Figures and Tables

**Figure 1 polymers-15-02347-f001:**
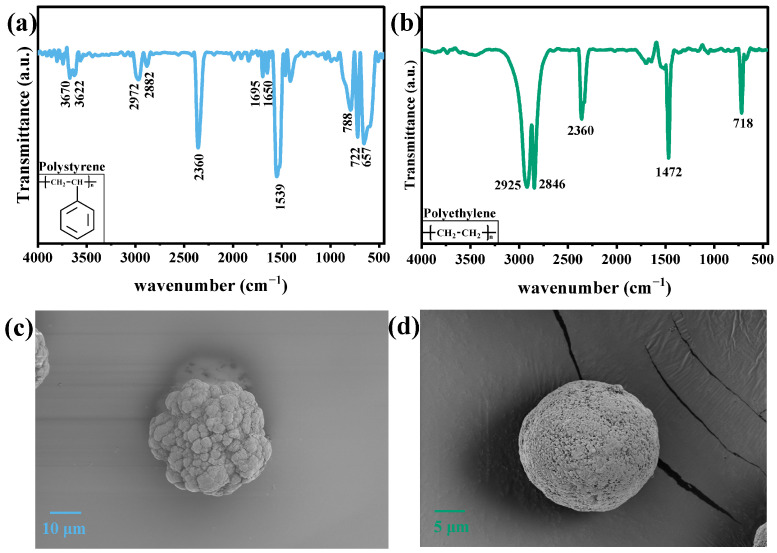
FTIR spectrum of (**a**) PS and (**b**) PE; SEM images of (**c**) PS and (**d**) PE.

**Figure 2 polymers-15-02347-f002:**
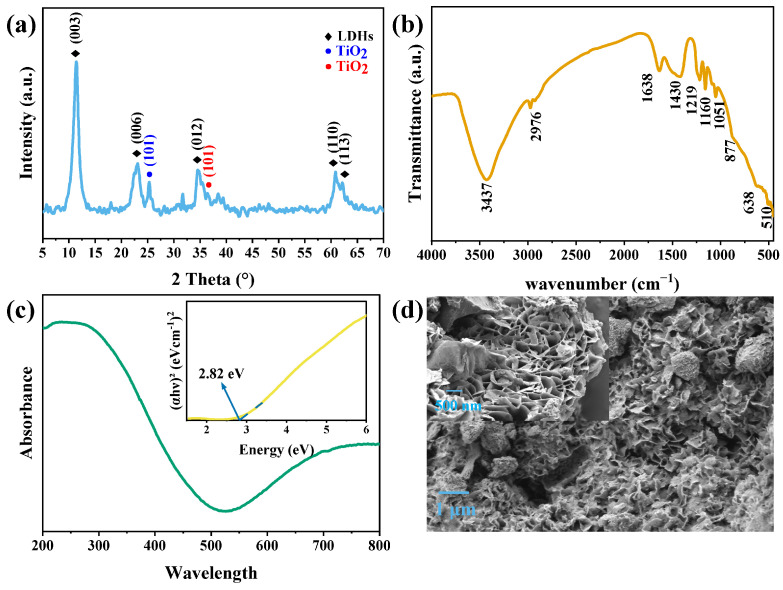
(**a**) XRD patterns, (**b**) FTIR spectra, (**c**) absorption spectra and Tauc plot, and (**d**) SEM image of CuMgAlTi-R400.

**Figure 3 polymers-15-02347-f003:**
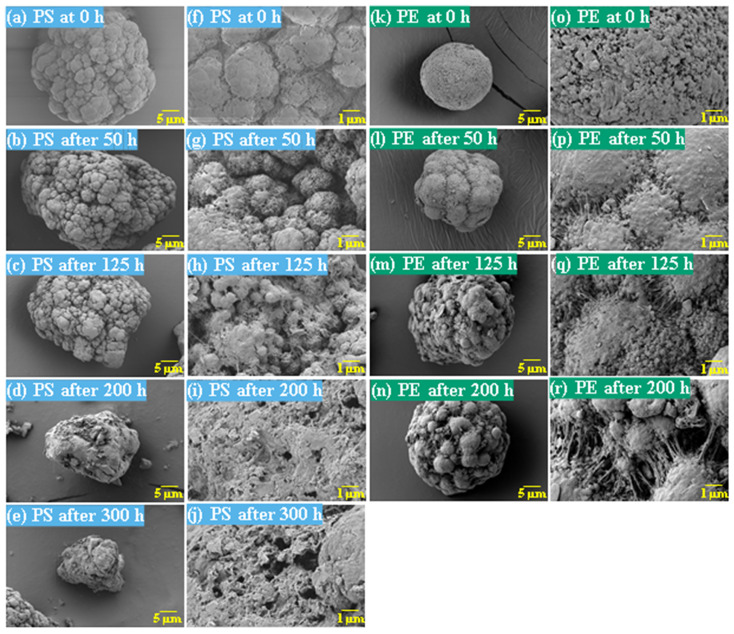
SEM images of PS (**a**–**j**) and PE (**k**–**r**).

**Figure 4 polymers-15-02347-f004:**
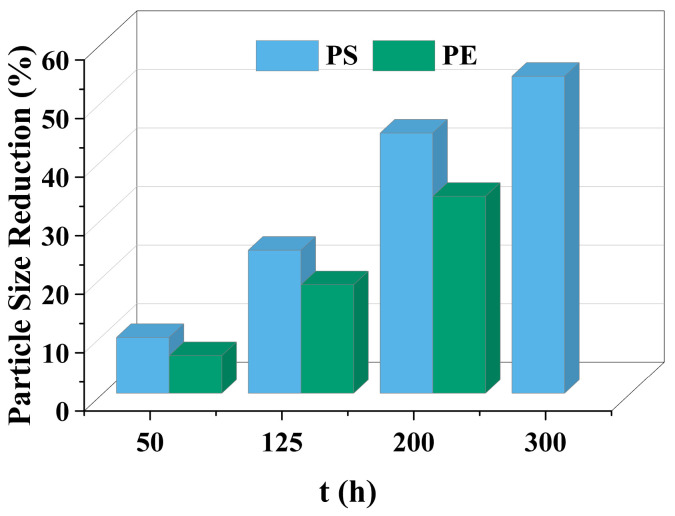
The size reduction percentage of microplastic particles as a function of exposure time.

**Figure 5 polymers-15-02347-f005:**
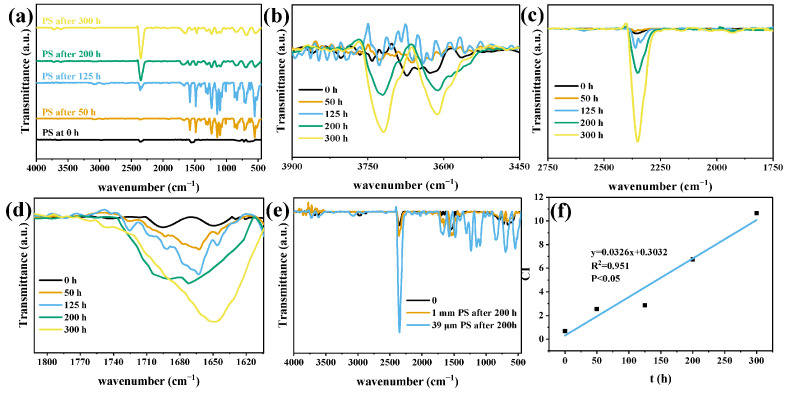
FTIR analysis of PS before and after degradation processes depicting (**a**) the overview spectra, (**b**) zoom of the region between 3900 and 3450 cm^−1^, (**c**) zoom of the region between 2750 and 1750 cm^−1^, and (**d**) zoom of the region between 1800 and 1600 cm^−1^; (**e**) FTIR spectrum of PS 1 mm and 39 μm and (**f**) correlation between CI and irradiation time.

**Figure 6 polymers-15-02347-f006:**
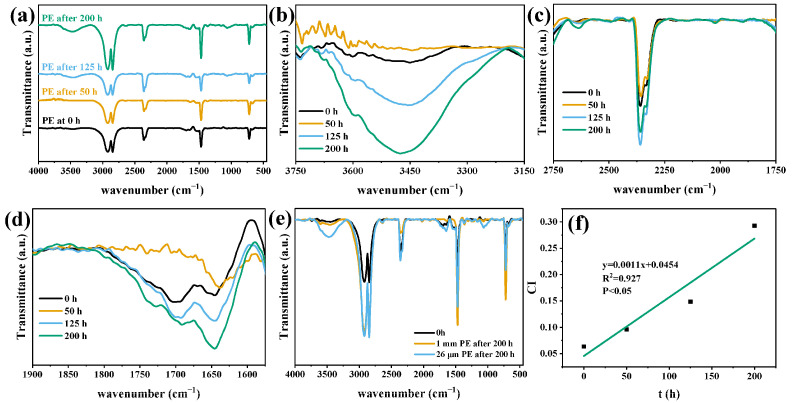
FTIR analysis of PE before and after degradation processes depicting (**a**) the overview spectra, (**b**) zoom of the region between 3750 and 3150 cm^−1^, (**c**) zoom of the region between 2750 and 1750 cm^−1^, and (**d**) zoom of the region between 1900 and 1600 cm^−1^; (**e**) FTIR spectrum of PE 1 mm and 26 μm; (**f**) correlation between CI and irradiation time.

**Figure 7 polymers-15-02347-f007:**
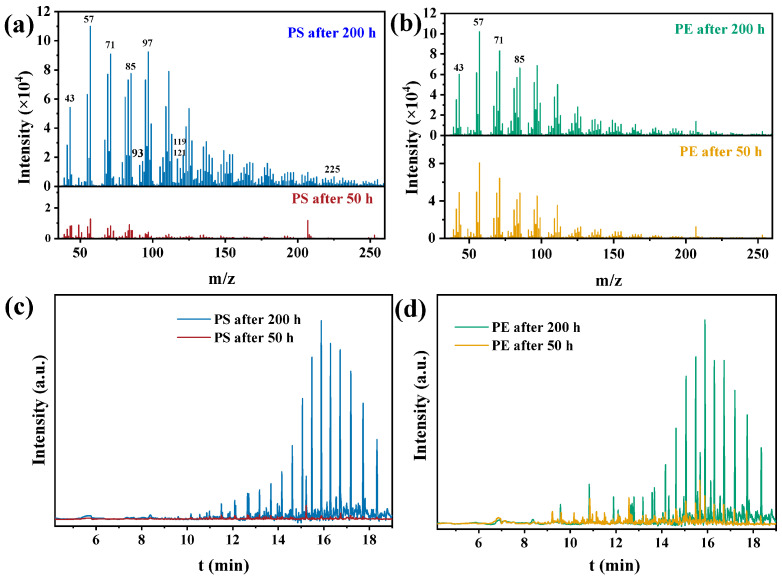
GC/MS spectrum of PS (**a**,**c**) and PE (**b**,**d**).

**Figure 8 polymers-15-02347-f008:**
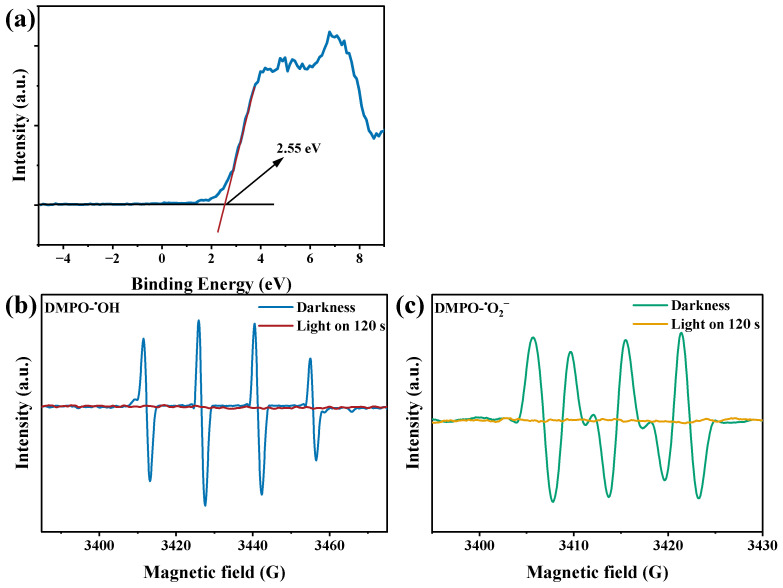
(**a**) VB-XPS curves of CuMgAlTi-R400, (**b**) EPR spectrum of DMPO- ^•^OH, (**c**) EPR spectrum of DMPO-^•^O_2_^−^.

**Figure 9 polymers-15-02347-f009:**
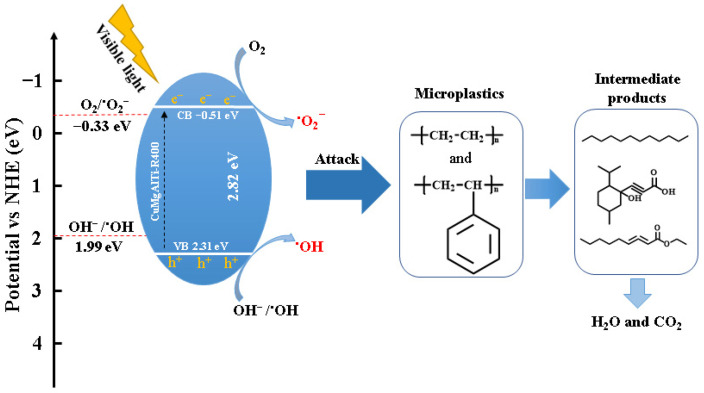
The probable mechanism responsible for the degradation of MPs by CuMgAlTi-R400 under visible light irradiation.

**Table 1 polymers-15-02347-t001:** Changes in PS microplastic particle size as a function of photocatalytic treatment for different periods of time.

Exposure Time (h)	Particle Size (μm)	Particle Size Distribution	Optical Micrograph Image
0	39.0 ± 5.6	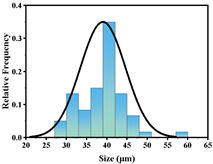	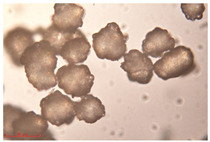
50	35.3 ± 5.0	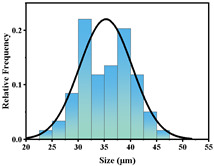	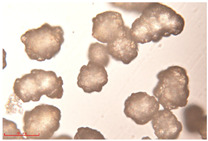
125	29.5 ± 6.1	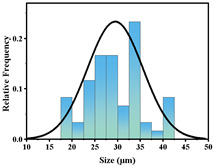	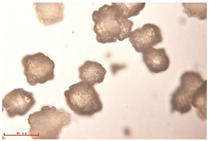
200	21.7 ± 7.1	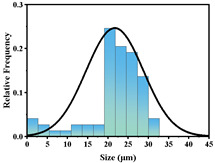	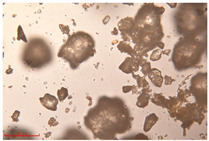
300	17.9 ± 5.9	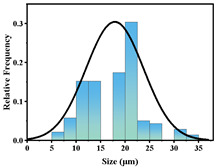	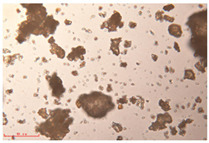

**Table 2 polymers-15-02347-t002:** Changes in PE microplastic particle size as a function of photocatalytic treatment for different periods of time.

Exposure Time (h)	Particle Size (μm)	Particle Size Distribution	Optical Micrograph Image
0	25.9 ± 3.6	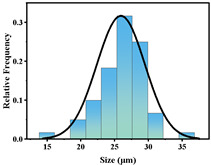	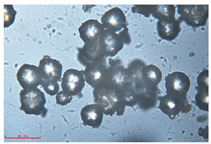
50	24.2 ± 4.8	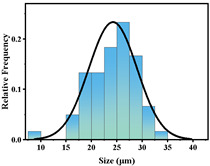	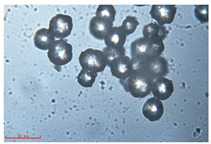
125	21.1 ± 6.0	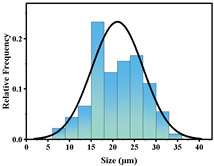	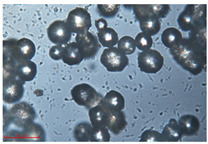
200	17.2 ± 7.2	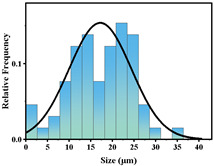	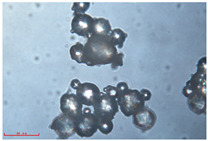

**Table 3 polymers-15-02347-t003:** Comparison of photocatalytic properties of the as-prepared samples with published papers.

Photocatalysts	Microplastic Type	Reaction Conditions	Removal Efficiency	Ref.
ZnO	PP (154.8 ± 1.4 µm)	120 W (ES- HALOGEN) (60 mW/cm^2^), Reaction time = 456 h	65%	[37]
N/TiO_2_	PE (382 ± 154 µm)	50 W visible LED lamp (400–800 nm); Reaction time = 50 h	4.7%	[38]
BiOCl-X	PE (200–250 μm)	250 W Xe lamp (λ > 420 nm), Reaction time = 5 h	5.38%	[39]
CuMgAlTi-R400	PS (39.0 ± 5.6μm)	50 W Xe lamp (λ = 320–780 nm), Reaction time = 300 h	54.2%	This work
CuMgAlTi-R400	PE (25.9 ± 3.6μm)	50 W Xe lamp (λ = 320–780 nm), Reaction time = 200 h	33.7%	This work

**Table 4 polymers-15-02347-t004:** Intermediate products identified by GC/MS during the photocatalytic degradation of PS.

Compound	Structure	Molecular Formula	Molecular Mass
2,4a,8,8-Tetramethyldecahydrocyclopropa[d]naphthalen	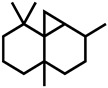	C_15_H_26_	206.2
2-Dodecen-1-ylsuccinic anhydride	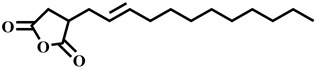	C_16_H_26_O_3_	266.2
Nona-2,3-dienoic acid, ethyl ester	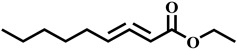	C_11_H_18_O_2_	182.1
Ppropiolic acid, 3-(1-hydroxy-2-isopropyl-5-methylcyclohexyl)-	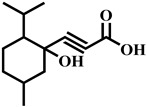	C_13_H_20_O_3_	224.1
2-Isopropyl-5-methylhex-2-enal	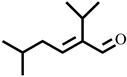	C_10_H_18_O	154.1

**Table 5 polymers-15-02347-t005:** Intermediate products identified by GC/MS during the photocatalytic degradation of PE.

Compound	Structure	Molecular Formula	Molecular Mass
Cyclopropane, (1-methyl-1,2-propadienyl)-	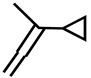	C_7_H_10_	94.1
Dodecane, 2,6,10-trimethyl-	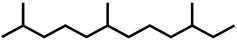	C_15_H_32_	212.2
1-Octanol, 2-butyl-	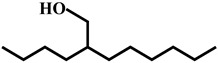	C_12_H_26_O	186.2
2-Pentanone, 5-(1,2-propadienyloxy)-	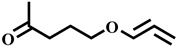	C_8_H_12_O_2_	140.1
Dodecane	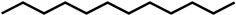	C_12_H_26_	170.2

## Data Availability

The data supporting reported results can be received from the authors.

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
