# Peer review of "Novel CuMgAlTi-LDH Photocatalyst for Efficient Degradation of Microplastics under Visible Light Irradiation"

_polymers, 2023, doi:10.3390/polym15102347_

Round 1
Reviewer 1 Report
The following points should be modified before the acceptance
1. The authors should mention the chemical concentration in moles or M instead of ratio in the section 2.2.
2. The photocatalytic reaction condition and concentration of both catalysts and MP should be mentioned in section 2.3
3. Authors should draw the reaction mechanism
4. The reaction conditions and degradation pathway should be demonstrated in the revised manuscript with proper band potential of the catalyst
5. Authors should mention the required band potential for MP degradation and their reaction conditions should be mentioned
Reviewer 2 Report
I believe that the manuscript can be accepted for publication in polymers after accomplishing the following major revisions:
Introduction. Add the reactions corresponding to the photocatalysis processes.
Introduction. The authors could mention/explore other advanced oxidation technology used successfully for water remediation.
Experimental. The irradiance and/or photon flux of the UV lamp must be added.
Results. The mechanism is not carefully investigated. EPR test results of free radicals need to be provided to provide direct evidence of free radical production.
Results. A comprehensive benchmarking of the material against others is needed.
Round 2
Reviewer 2 Report
The authors have made considerable effort to improve the work. Therefore, I recommend the publication of the manuscript.